The non-avian theropod quadrate I: standardized terminology with an overview of the anatomy and function

Hendrickx Christophe 1 2 6 christophe.hendrickx@hotmail.com
Araújo Ricardo 2 3 4 5
Mateus Octávio 1 2
1 Departamento de Ciências da Terra, Universidade Nova de Lisboa, GeoBioTec, Faculdade de Ciências e Tecnologia , Caparica , Portugal
2 Museu da Lourinhã , Lourinhã , Portugal
3 Huffington Department of Earth Sciences, Southern Methodist University , Dallas, TX , USA
4 Instituto Superior Técnico, Universidade de Lisboa , Lisboa , Portugal
5 Museum für Naturkunde , Berlin , Germany
6 Current affiliation: Evolutionary Studies Institute, Center of Excellence in Palaeosciences, University of the Witwatersrand , South Africa
Farke Andrew
Electronic publication date: 2015 Sep 17
Publication date: 2015
Volume: 3
Electronic Location ID: e1245
Received 2014 May 4; Accepted 2015 Aug 25
Copyright: © 2015 Hendrickx et al.
Copyright year: 2015
Copyright holder: Hendrickx et al.
License: This is an open access article distributed under the terms of the Creative Commons Attribution License, which permits unrestricted use, distribution, reproduction and adaptation in any medium and for any purpose provided that it is properly attributed. For attribution, the original author(s), title, publication source (PeerJ) and either DOI or URL of the article must be cited.
License URL: https://creativecommons.org/licenses/by/4.0/

Keywords: Quadrate, Terminology, Anatomy, Theropod, Dinosaur, Mandibular articulation

Funding: Fundação para a Ciência e a Tecnologia (FCT) SFRH/BD/62979/2009 SFRH/BPD/96205/2013 This research was supported by the Fundação para a Ciência e a Tecnologia (FCT) scholarships (Ministério da Ciência, Tecnologia e Ensino superior, Portugal) SFRH/BD/62979/2009 (CH) and SFRH/BPD/96205/2013 (RA). The funders had no role in study design, data collection and analysis, decision to publish, or preparation of the manuscript.

==============================
The quadrate of reptiles and most other tetrapods plays an important morphofunctional role by allowing the articulation of the mandible with the cranium. In Theropoda, the morphology of the quadrate is particularly complex and varies importantly among different clades of non-avian theropods, therefore conferring a strong taxonomic potential. Inconsistencies in the notation and terminology used in discussions of the theropod quadrate anatomy have been noticed, including at least one instance when no less than eight different terms were given to the same structure. A standardized list of terms and notations for each quadrate anatomical entity is proposed here, with the goal of facilitating future descriptions of this important cranial bone. In addition, an overview of the literature on quadrate function and pneumaticity in non-avian theropods is presented, along with a discussion of the inferences that could be made from this research. Specifically, the quadrate of the large majority of non-avian theropods is akinetic but the diagonally oriented intercondylar sulcus of the mandibular articulation allowed both rami of the mandible to move laterally when opening the mouth in many of theropods. Pneumaticity of the quadrate is also present in most averostran clades and the pneumatic chamber—invaded by the quadrate diverticulum of the mandibular arch pneumatic system—was connected to one or several pneumatic foramina on the medial, lateral, posterior, anterior or ventral sides of the quadrate.

Introduction

The quadrate (in Latin quadratum, meaning ‘square’) is a cranial bone of endochondral origin that articulates with the mandible in all gnathostomes except mammaliaforms, which have had the quadrate evolved into the incus (Reichert, 1837; Takechi & Kuratani, 2010; Brusatte, 2012; Benton, 2015). In theropods, this bone plays many important functions such as a structural support for the basicranium, articulatory element with the lower jaws, attachment for several muscles, hearing, and hosting important nerves, pneumatic sinuses, and vascular passages (e.g., Witmer, 1990; Witmer, 1997; Bakker, 1998; Sedlmayr, 2002; Kundrát & Janáček, 2007; Holliday & Witmer, 2008; Tahara & Larsson, 2011; see Appendix S1).

Although the outward morphology of the quadrate is relatively simple, it varies significantly among theropods in the structure of its head, mandibular articulation, quadratojugal contact and the presence of pneumatic openings, quadrate foramen, and lateral process (e.g., Holtz, 2003; Therrien, Henderson & Ruff, 2005; Hone & Rauhut, 2010; Zanno & Makovicky, 2011). Variation in the quadrate morphology in the derived theropod group Aves has long been used as a mean of systematic significance (e.g., Lowe, 1926; Samejima & Otsuka, 1987; Barbosa, 1990; Elzanowski, Paul & Stidham, 2001; Elzanowski & Stidham, 2010). Similarly, but to a lesser degree, the systematic potential of the quadrate bone has also been noted for non-avian theropods (Maryańska & Osmólska, 1997; Currie, 2006), highlighting the importance that should be given to the description of this bone in the literature on non-avian theropod anatomy. Nevertheless, the terminology and abbreviations of the quadrate anatomy has been inconsistent in non-avian theropods, and several different anatomical terms for the same quadrate sub-entity are often used (see Appendix S2). Although a list of anatomical terms has been given by Baumel & Witmer (1993), Elzanowski, Paul & Stidham (2001) and Elzanowski & Stidham (2010) for the avian quadrate, the terminology proposed by these authors has not been applied to the description of the non-avian theropod quadrate hitherto. Indeed, the quadrate of birds has greatly changed in its morphology throughout the evolution of this clade and hence displays many features absent in more primitive theropods. Thus, many anatomical terms coined by Baumel & Witmer (1993), Elzanowski, Paul & Stidham (2001) and Elzanowski & Stidham (2010) cannot be applied to the non-avian theropod quadrate. Moreover, some quadrate entities such as the quadrate foramen and the lateral process observable in non-avian theropods are absent in their avian descendants and do not appear in the list made by these authors.

The work presented here has two major aims. First, we propose a standardization of the anatomical terms for the quadrate sub-units, each associated with a two to four letters abbreviation and followed by a definition, in order to facilitate future descriptions of this bone in the literature. Second, we present and discuss the current knowledge on the function and pneumaticity of this important bone in non-avian theropods. A comprehensive study on the anatomy and phylogenetic potential of the non-avian theropod quadrate through cladistic and phylogenetic morphometric analyses will be provided in a companion article that will be published later.

Theropod classification

The theropod phylogeny adopted here follows the classification summarized by Hendrickx, Hartman & Mateus (2015) for non-avian theropods. Megaraptoran theropods are here considered as belonging to the clade Allosauroidea, as proposed by Benson, Carrano & Brusatte (2010) and Carrano, Benson & Sampson (2012). The phylogenetic definitions of each non-avian theropod clade also follow those compiled and given by Hendrickx, Hartman & Mateus (2015): Table 1.

Proposed Terminology of the Quadrate Anatomy

Favored terminology

The anatomical terms of the theropod quadrate were grouped in five main sections: quadrate body, quadrate head, mandibular articulation, pterygoid flange, and pneumatic openings. The terms for each quadrate sub-units were selected by their relevance, significance and importance in the non-avian theropod literature. The non-standardized traditional Romerian directional and anatomical terms (Romer, 1956; Wilson, 2006) were, therefore, favored over the terminology of the Nomina Anatomica Veterinaria (NAV) updated by the ICVGAN (2012) and the Nomina Anatomica Avium (NAA) provided by Baumel (1993) as Romerian terms are the most commonly used in the non-avian theropod literature (e.g., Eddy & Clarke, 2011; C Hendrickx, pers. obs., 2015). Consequently, ‘anterior’ and ‘posterior’ are used as directional terms in lieu of the veterinarian alternatives ‘cranial’ and ‘caudal,’ respectively. Because non-avian archosaurs are the direct ancestors of birds, Harris (2004) recommended to adopt the NAA as the standardized nomenclature to describe all archosaurs (and even diapsids), yet we favor Wilson’s (2006) opinion to retain Romerian terms for non-avian dinosaurs. As noted by Wilson (2006), the Romerian nomenclature is the lingua franca for most of the dinosaur/archosaur literature. In addition, standard terminologies using Romerian terms are often proposed to describe the saurischian anatomy (e.g., Wilson, 1999; Wilson et al., 2011; Hendrickx & Mateus, 2014; Hendrickx, Mateus & Araújo, 2015). Comparison between the NAA nomenclature and the Romerian terminology here proposed for the quadrate anatomy is provided in Fig. 1 and Table 1.

Table 1 Proposed terminology and abbreviations of the non-avian theropod quadrate.

Standardized terminology and abbreviations of the non-avian theropod quadrate and comparison with the terminology of the avian quadrate based on Baumel & Witmer (1993), Elzanowski, Paul & Stidham (2001) and Elzanowski & Stidham (2010).

Non-avian theropod quadrate		Avian theropod quadrate	
Quadrate	q	Os quadratum (Quadratum)	
Quadrate body	qb	Corpus quadrati	
Quadrate shaft	qs	/	
Quadrate ridge	qr	/	
Quadrate ridge groove	qrg	/	
Quadrate foramen	qf	/	
Lateral process	lpq	/	
Quadratojugal contact	qjc	Cotyla quadratojugalis	
Ventral quadratojugal contact	vqjc	/	
Dorsal quadratojugal contact	dqjc	/	
Quadratojugal process	qjp	/	
Ventral projection of the dorsal quadratojugal contact	vpdq	/	
Dorsal projection of the ventral quadratojugal contact	dpvq	/	
Squamosal contact	sqc	/	
Posterior fossa	pfq	/	
Quadrate head	qh	Caput quadrati	
Otic capitulum	oca	Capitulum oticum	
Squamosal capitulum	sca	Capitulum squamosum	
Intercapitular sulcus	icas	Incisura/Vallecula intercapitularis	
Mandibular articulation	mar	Pars/Processus mandibularis	
Ectocondyle	ecc	Condylus (mandibularis) lateralis	
Entocondyle	enc	Condylus (mandibularis) medialis	
Mediocondyle	mec	Condylus caudalis	
Intercondylar sulcus	ics	Sulcus/Vallecula intercondylaris	
Anterior intercondylar notch	ain	/	
Posterior intercondylar notch	pin	/	
Pterygoid flange	pfl	Processus orbitalis	
Pterygoid contact	ptc	Condylus pterygoideus/Facies articularis pterygoidea	
Medial fossa	mfq	Fossa basiorbitalis	
Ventral shelf	vsh	/	
Quadrate pneumatic foramen	qpf	/	
Dorsal pneumatic foramen	dpf	/	
Medial pneumatic foramen	mpf	Foramen pneumaticum basiorbitale	
Posterior pneumatic foramen	ppf	Foramen pneumaticum caudomediale	
Anterior pneumatic foramen	apf	Foramen pneumaticum rostromedial	
Ventral pneumatic foramen	vpf	Foramen pneumaticum adventitium	
Posterior pneumatic fossa	ppfo	/	
Lateral pneumatic fossa	lpfo	/	

Romer’s (1956) terminology of the quadrate is limited. He only expanded the vocabulary to describe this bone in reptiles to six terms, namely: the main body, quadrate shaft, quadrate foramen, quadrate head, quadrate flange and articular termination. Three terms were kept as such in the proposed terminology of the quadrate (i.e., quadrate shaft, quadrate foramen, and quadrate head) and the three others were slightly modified. The quadrate body (instead of “main body of [the] quadrate” sensu Romer, 1956: p. 640), mandibular articulation (instead of “articular termination” sensu Romer, 1956: p. 632) and pterygoid flange (instead of “quadrate flange” sensu Romer, 1956: p. 146) were chosen not only because they are more commonly used in the theropod literature currently describing the quadrate (C Hendrickx, pers. obs., 2015), but are also more specific of the loci of the anatomical sub-entity described. It should be noted that the pterygoid flange of Romer (1956) describes a wing-like process of the pterygoid and not the anteriorly projected ramus of the quadrate.

Quadrate body

Quadrate body (qb)

Part of the quadrate that includes the quadrate shaft, the quadrate ridge, the lateral contact (quadratojugal and/or squamosal contact), and the lateral process, and excludes the quadrate head, mandibular articulation, and pterygoid flange (Figs. 1G and 2A). In posterior view, the quadrate body is delimited by the lateral margin of the lateral contact and sometimes by the medial margin of the quadrate foramen, the dorsal margin of the mandibular articulation, the ventral margin of the quadrate head, and a medial margin mostly formed by the quadrate shaft and the medial fossa of the pterygoid flange. The quadrate body is equivalent to the ‘Corpus ossis quadrati’ of Baumel & Witmer (1993), and the ‘Corpus quadrati’ of Elzanowski, Paul & Stidham (2001) and Elzanowski & Stidham (2010) for avian theropods (Fig. 1A).

Figure 1 Avian and non-avian theropod terminology of the quadrate bone.

Left quadrate of the common ostrich Struthio camelus (NH.11.75; courtesy of Paolo Viscardi) in (A, G) anterior, (B, H) lateral, (C, I) posterior, (D, J) medial, (E, K) dorsal, and (F, L) ventral views. The ostrich quadrate is annotated with (A–F) Baumel & Witmer (1993), Elzanowski, Paul & Stidham (2001) and Elzanowski & Stidham’s (2010) terminologies, and (G–L) the here proposed terminology for the non-avian theropod quadrate.

Figure 2 Anatomy of non-avian theropod quadrates.

Line drawings of the right (A–E) quadrate of Tsaagan mangas (IGM 100-1015) in (A) anterior, (B) lateral, (C) posterior, (D) medial and (E) ventral views; left (F–I) and right (J–K) quadrates (F) of Baryonyx walkeri (NHM R9951), (G) Aerosteon riocoloradensis (MCNA-PV-3137), (H) an indeterminate Oviraptoridae (IGM A; Maryańska & Osmólska, 1997), (I) Tyrannosaurus rex (BHI 3333; Larson & Carpenter, 2008), (J) Allosaurus ‘jimmadseni’ (SMA 0005), and (K) Majungasaurus crenatissimus (FMNH PR 2100) in (F–I) posterior and (J–K) ventral views. Abbreviations: ain, anterior intercondylar notch; dqjc, dorsal quadratojugal contact; ecc, ectocondyle; enc, entocondyle; ics, intercondylar sulcus; lpq, lateral process of the quadrate; mar, mandibular articulation; mfq, medial fossa of the quadrate; oca, otic capitulum; pfl, pterygoid flange; pfq, posterior fossa of the quadrate; pin, posterior intercondylar notch; ppf, posterior pneumatic foramen; qb, quadrate body; qf, quadrate foramen; qh, quadrate head; qj, quadratojugal; qjp, quadratojugal process; qr, quadrate ridge; qrg, quadrate ridge groove; qs, quadrate shaft; sqc, squamosal contact; sca, squamosal capitulum; vqjc, ventral quadratojugal contact; vpdq, ventral projection of the dorsal quadratojugal contact; vsh, ventral shelf.

Quadrate shaft (qs)

Part of the quadrate body that excludes the lateral process and all articulating surfaces (i.e., quadrate head, quadratojugal/squamosal/pterygoid contacts, and mandibular articulation; Fig. 2C). The quadrate shaft, as called by Welles (1984), Sereno & Novas (1994), Norell et al. (2006), Sampson & Witmer (2007), Sereno et al. (2008), Carrano, Loewen & Sertich (2011), Brusatte, Carr & Norell (2012), Choiniere et al. (2014a), and Choiniere et al. (2014b), is also referred as the ‘quadrate pillar’ by Madsen & Welles (2000), and the ‘ascending process’ by Colbert (1989).

Quadrate ridge (qr)

Ventrodorsally elongated column, ridge or crest located on the quadrate body and visible in posterior view (Figs. 2C, 2F–2K). Although the quadrate ridge is present in the large majority of non-avian theropods, a description of the structure is often omitted in the literature. The quadrate ridge is referred as a ‘column’ by Welles (1984), a ‘ridge-like mediodorsal edge’ by Carr (1996), a ‘prominent rounded ridge’ by Smith et al. (2007), a ‘columnar ridge’ by Rauhut, Milner & Moore-Fay (2010), a ‘robust ridge’ by Brusatte, Carr & Norell (2012), a ‘ridge’ or ‘pillar’ by Choiniere et al. (2014a), and a ‘bulging ridge’ by Lautenschlager et al. (2014).

Quadrate ridge groove (qrg)

Groove dividing the quadrate ridge in two different units at two-thirds, or more dorsally, of the quadrate height (Fig. 2G). A quadrate ridge groove exists in some allosauroid theropods.

Quadrate foramen (qf)

Aperture in the quadrate body or concavity on the lateral margin of the quadrate body and delimited ventrally by the ventral quadratojugal contact and dorsally by the dorsal quadratojugal contact and its ventral projection in some theropod taxa (Figs. 2A, 2E–2G and 2I). Most authors usually refer to this perforation as the quadrate foramen (e.g., Welles, 1984; Sereno & Novas, 1994; Charig & Milner , 1997; Maryańska & Osmólska, 1997; Currie & Carpenter, 2000; Coria & Currie, 2006; Currie, 2006; Norell et al., 2006; Choiniere et al., 2010; Choiniere et al., 2014a; Choiniere et al., 2014b; Zanno, 2010; Brusatte, Carr & Norell, 2012). Yet, it can be also called the ‘paraquadratic foramen’ (e.g., Barsbold & Osmólska, 1999; Kobayashi & Lü, 2003; Kobayashi & Barsbold, 2005), the ‘paraquadrate foramen’ (Sampson & Witmer, 2007; Dal Sasso & Maganuco, 2011; Lautenschlager et al., 2014), the ‘paraquadrate fenestra’ (Smith et al., 2007) or the ‘quadrate fenestra’ (e.g., Carr , 1996; Sereno et al., 1998; Currie, 2003; Eddy & Clarke, 2011). A quadrate foramen exists in all non-avian theropods but Ceratosauria and Megalosauridae.

Lateral process (lpq).

Lateral or anterolateral projection of the lateral margin of the quadrate body (Fig. 2B). Also known as the ‘dorsal wing’ (Welles, 1984; Currie, 2006), the ‘anterolateral wing’ (Madsen & Welles, 2000), the ‘lateral lamina’ (Coria & Salgado, 1998) and the ‘lateral ramus’ (Sampson & Witmer, 2007), this process can contact the quadratojugal and/or the squamosal and therefore either be referred to the ‘quadratojugal ramus’ (Sampson & Witmer, 2007) or the ‘squamosal ramus’ (Norell et al., 2006).

Quadratojugal contact (qjc)

Area of contact of the quadrate with the quadratojugal on the lateral, anterolateral, or posterolateral margin of the quadrate body (Fig. 2G). The quadratojugal contact, which is similar to the ‘cotyla quadratojugalis’ of Baumel & Witmer (1993), Elzanowski, Paul & Stidham (2001) and Elzanowski & Stidham (2010) for avian theropods (Fig. 1B), can be divided into a ventral and a dorsal quadratojugal contact when the quadrate foramen is present and delimited by both quadrate and quadratojugal.

Ventral quadratojugal contact (vqjc)

Ventral area of contact of the quadrate with the quadratojugal (Figs. 2B, 2F and 2H). The ventral quadratojugal contact of the quadrate always receives the quadratojugal bone.

Dorsal quadratojugal contact (dqjc)

Dorsal area of contact of the quadrate with the quadratojugal (Figs. 2B and 2F). The ventral quadratojugal contact of the quadrate can either receive the quadratojugal or both quadratojugal and squamosal in some theropod taxa.

Ventral projection of the dorsal quadratojugal contact (vpdq)

Small projection of the dorsal quadratojugal contact delimiting the laterodorsal margin of the quadrate foramen (Fig. 2I).

Dorsal projection of the ventral quadratojugal contact (dpvq)

Small projection of the ventral quadratojugal contact delimiting the lateroventral margin of the quadrate foramen.

Quadratojugal process (qjp)

Anterior projection of the ventral quadratojugal contact of the quadrate (Fig. 2B). Also known as the ‘quadratojugal lamina’ (Lautenschlager et al., 2014).

Lateroventral process (lvp)

Lateromedially oriented ventral projection of the ventral quadratojugal contact of the quadrate that bounds the quadratojugal ventrally (Fig. 2H). The lateroventral process is similar to the ‘lateral process’ of Maryańska & Osmólska (1997).

Squamosal contact (sqc)

Contact on the lateral margin of the quadrate with the squamosal (Fig. 2B).

Posterior fossa (pfq)

Depression or concavity situated on the posterior side of the quadrate body and dorsal to the mandibular articulation, ventral to the quadrate head and lateral to the quadrate ridge (Fig. 2B). The posterior fossa can include or exclude the quadrate foramen.

Quadrate head

Quadrate head (qh)

Dorsal articulation of the quadrate abutting to the cotyle of the squamosal and contacting other bones of the braincase in some theropod taxa (Fig. 2D). The quadrate head, as it is called by Britt (1991), Charig & Milner (1997), Madsen & Welles (2000), Sampson & Witmer (2007), Sereno et al. (2008), Norell et al. (2009), Brusatte, Carr & Norell (2012), Choiniere et al. (2014a), Choiniere et al. (2014b) and Lautenschlager et al. (2014) among others, has also been termed ‘quadrate cotylus’ (Currie, 2003; Coria & Currie, 2006), ‘quadrate cotyle’ (Currie, 2003; Coria & Currie, 2006), ‘squamosal condyle’ (Coria & Salgado, 1998), ‘squamosal articulation’ (Turner, Pol & Norell, 2011), ‘dorsal articular surface’ (Larson, 2013), and ‘otic process’ (Maryańska & Osmólska, 1997; Burnham, 2004; Holliday & Witmer, 2008). In avian theropods, the quadrate head is homologous to the ‘Caput quadrati’ of Elzanowski, Paul & Stidham (2001) and Elzanowski & Stidham (2010), and roughly equivalent to the ‘Processus oticus’ (Baumel & Witmer, 1993). In birds, the ‘Processus oticus’ (Baumel & Witmer, 1993), and the ‘Pars oticus’ of Elzanowski, Paul & Stidham (2001) and Elzanowski & Stidham (2010) also includes several sub-units that are either absent in non-avian theropods (e.g., ‘Crista Tympanica’, ‘Tuberculum subcapitulare’), or here included in the quadrate body (e.g., ‘Sulcus pneumaticus’, ‘Foramen pneumaticum rostromediale’). The bistylic quadrate head present in some tyrannosaurids, alvarezsauroids, oviraptorids and avian theropods is divided into otic and squamosal capitula.

Otic capitulum (oca)

Medial capitulum of the quadrate head articulating with the braincase (Fig. 2H). The otic capitulum is referred as the ‘capitulum (condylus) oticum’ by Baumel & Witmer (1993), Elzanowski, Paul & Stidham (2001) and Elzanowski & Stidham (2010) for avian theropods (Fig. 1A).

Squamosal capitulum (sca)

Lateral capitulum of the quadrate head articulating with the squamosal (Fig. 2H). The squamosal capitulum is similar to the ‘capitulum (condylus) squamosum’ of Baumel & Witmer (1993), Elzanowski, Paul & Stidham (2001) and Elzanowski & Stidham (2010) for avian theropods (Fig. 1C).

Intercapitular sulcus (icas)

Groove separating the ootic capitulum from the squamosal capitulum on the dorsal surface of the quadrate head (Fig. 2H). The intercapitular sulcus (Witmer, 1990) is equivalent to the ‘incisura intercapitularis’ of Baumel & Witmer (1993), and the ‘vallecula intercapitularis’ of Elzanowski, Paul & Stidham (2001) and Elzanowski & Stidham (2010) for avian theropods (Fig. 1E).

Mandibular articulation

Mandibular articulation (mar)

Ventral surface of the quadrate, articulating with the mandible and fitting into the glenoid fossa of the lower jaw. It includes the ectocondyle, entocondyles, sometimes a mediocondyle, and a single intercondylar sulcus, even when three condyles are present (Fig. 2C). The mandibular articulation, also known as the ‘mandibular capitulum’ (Lautenschlager et al., 2014), is equivalent to the ‘Processus mandibularis’ of Baumel & Witmer (1993), and the ‘Pars mandibularis’ of Elzanowski, Paul & Stidham (2001) and Elzanowski & Stidham (2010) for avian theropods (Fig. 1A). Although most authors (e.g., Currie, 2006; Sampson & Witmer, 2007; Rauhut, Milner & Moore-Fay, 2010; Brusatte, Carr & Norell, 2012; Lautenschlager et al., 2014) referred the ectocondyle and entocondyles as the lateral and medial condyles (or hemicondyles) respectively, the terms ectocondyle and entocondyle have been used by Welles (1984) and Madsen & Welles (2000). The condyle present in between the ecto- and entocondyles in some theropods is here coined mediocondyle.

Ectocondyle (ecc)

Lateral condyle of the mandibular articulation (Fig. 2). The ectocondyle is equivalent to the ‘condylus (mandibularis) lateralis’ of Baumel & Witmer (1993), Elzanowski, Paul & Stidham (2001) and Elzanowski & Stidham (2010) for avian theropods (Fig. 1F).

Entocondyle (enc)

Medial condyle of the mandibular articulation (Fig. 2). The entocondyle has been referred as the ‘condylus (mandibularis) medialis’ by Baumel & Witmer (1993), Elzanowski, Paul & Stidham (2001) and Elzanowski & Stidham (2010) for avian theropods (Fig. 1F).

Mediocondyle (mdc)

Posterior condyle of the mandibular articulation located between the ecto- and entocondyles. The mediocondyle is referred as the ‘third condyle’ by Clark, Perle & Norell (1994) and Xu & Wu (2001), the ‘accessory condyle’ by Kobayashi & Lü (2003) and Lautenschlager et al. (2014), and the ‘condylus caudalis’ of Baumel & Witmer (1993) and Elzanowski, Paul & Stidham (2001) for avian theropods.

Intercondylar sulcus (ics)

Groove separating the ectocondyle from the entocondyle and articulated with the interglenoid ridge of the articular (Figs. 2E and 2K). The intercondylar sulcus, a term also used by Carrano, Loewen & Sertich (2011), can be referred as a ‘groove’ (e.g., Madsen, 1976; Britt, 1991; Madsen & Welles, 2000; Currie, 2006), ‘swelling’ (Charig & Milner , 1997), ‘sulcus’ (e.g., Kobayashi & Lü, 2003; Norell et al., 2006; Sadleir, Barrett & Powell, 2008), ‘trochlea’ (Brochu, 2003; Brusatte et al., 2010), ‘trochlear surface’ (Brusatte et al., 2010; Brusatte, Carr & Norell, 2012), and ‘intercondylar bridge’ (Zanno, 2010). The intercondylar sulcus is similar to the ‘sulcus intercondylaris’ (Baumel & Witmer, 1993) and the ‘vallecula intercondylaris’ (Elzanowski, Paul & Stidham, 2001; Elzanowski & Stidham, 2010) of the quadrate of avian theropods (Fig. 1F).

Anterior intercondylar notch (ain)

Notch located between the ectocondyle and entocondyle, on the anterior margin of the mandibular articulation (Fig. 2K).

Posterior intercondylar notch (pin)

Notch located between the ectocondyle and entocondyle, on the posterior margin of the mandibular articulation, and being referred as the ‘pit’ by Bakker (1998) (Fig. 2J).

Pterygoid flange

Pterygoid flange (pfl)

Ventrodorsally elongated sheet-like process projecting anteriorly or anteromedially from the medial side of the anterior surface of the quadrate body to contact the pterygoid bone (Figs. 2A and 2D). The pterygoid flange, a term also used by Charig & Milner (1997), Brochu (2003), Currie (2006), Coria & Currie (2006), Rauhut, Milner & Moore-Fay (2010) and Lautenschlager et al. (2014), is also known as the ‘quadrate/anterior flange’ (e.g., Colbert, 1989; Norell et al., 2006; Brusatte et al., 2010; Brusatte, Carr & Norell, 2012), the ‘pterygoid ramus’ (e.g., Sereno & Novas, 1994; Sampson & Witmer, 2007; Choiniere et al., 2010; Choiniere et al., 2014a; Choiniere et al., 2014b), the ‘pterygoid wing’ (e.g., Welles, 1984; Madsen & Welles, 2000; Eddy & Clarke, 2011), the ‘pterygoid ala’ (e.g., Currie, 2003; Currie, 2006; Sadleir, Barrett & Powell, 2008; Dal Sasso & Maganuco, 2011), the ‘pterygoid process’ (Molnar, 1991; Carr , 1996; Sereno et al., 2008), the ‘optic wing’ (Balanoff & Norell, 2012), the ‘orbital process’ (Clark, Perle & Norell, 1994; Chiappe, Norell & Clark, 2002), and the ‘processus orbitalis’ (Baumel & Witmer, 1993; Elzanowski, Paul & Stidham, 2001; Elzanowski & Stidham, 2010) for avian theropods (Fig. 1B).

Pterygoid contact (ptc)

Area of contact with the pterygoid on the medial margin of the pterygoid flange or the quadrate body (Fig. 2D). In avian theropods, the pterygoid contact is homologous to the ‘facies pterygoidea’ in Elzanowski, Paul & Stidham (2001) and the ‘facies articularis pterygoidea’ in Elzanowski & Stidham (2010). It is also homologous to the ‘condylus pterygoideus,’ located on the quadrate body, in Baumel & Witmer (1993), Elzanowski, Paul & Stidham (2001), and Elzanowski & Stidham (2010; Fig. 1D).

Medial fossa (mfq)

Depression or concavity located on the medial surface of the pterygoid flange, typically on its posteroventral extremity (Fig. 2D). The medial fossa is delimited by the quadrate shaft and the ventral shelf in some theropod taxa. The medial fossa is similar to the ‘fossa corporis quadrati’ of Fuchs (1954), and the ‘fossa basiorbitalis’ of Elzanowski, Paul & Stidham (2001) and Elzanowski & Stidham (2010) for avian theropods (Fig. 1D).

Ventral shelf (vsh)

A medial or posteromedial fold of the ventral margin of the pterygoid flange (Figs. 3A, 3G and 3M). The term ‘shelf’ was employed by Sereno & Novas (1994) and ventral shelf was used by Sampson & Witmer (2007), Eddy & Clarke (2011) and Carrano, Loewen & Sertich (2011).

Figure 3 Topological homologies in the non-averostran theropod quadrate.

Left (A, C, F) and right (B, D, E; reversed) quadrates of Dilophosaurus wetherilli (UCMP 37302) in (A) anterior, (B) lateral, (C) posterior, (D) medial, (E) dorsal and (F) ventral views (courtesy of Randall Irmis and Matthew Carrano). Right quadrate (G–L; reversed) of Majungasaurus crenatissimus (FMNH PR 2100) in (G) anterior, (H) lateral, (I) posterior, (J) medial, (K) dorsal, and (L) ventral views. Left quadrate (M–R) of Baryonyx walkeri (NHM R9951) in (M) anterior, (N) lateral, (O) posterior, (P) medial, (Q) dorsal, and (R) ventral views. Right quadrate (S–W) of Eustreptospondylus oxoniensis (OUMNH J.13558; reversed) in (S) anterior, (T) lateral, (U) posterior, (V) medial and (W) ventral views (courtesy of Paul Barrett). Abbreviations: afq, anterior fossa; ain, anterior intercondylar notch; dqjc, dorsal quadratojugal contact; ecc, ectocondyle; enc, entocondyle; ics, intercondylar sulcus; lpq, lateral process; mfq, medial fossa; pfq, posterior fossa; pfl, pterygoid flange; qf, quadrate foramen; qh, quadrate head; qjp, quadratojugal process; qr, quadrate ridge; vpdq, ventral projection of the dorsal quadratojugal contact; vqjc, ventral quadratojugal contact; vsh, ventral shelf of the pterygoid flange.

Pneumatic foramina and fossae

Quadrate pneumatic chamber (qpc)

Internal chamber within the quadrate, either fully contained within the bone or communicating externally by one or several pneumatic foramina. The quadrate pneumatic chamber hosts the quadrate sinus/diverticulum and, in some cases, includes several interconnected chambers separated by thin bony lamellae within the quadrate body and pterygoid flange (Kundrát & Janáček, 2007; Tahara & Larsson, 2011; Gold, Brusatte & Norell, 2013).

Dorsal pneumatic foramen (dpf)

Aperture located on the anterodorsal surface of the quadrate, just ventral to the quadrate head.

Medial pneumatic foramen (mpf)

Aperture or recess situated on the medial side of the quadrate, typically in the ventral portion of the medial surface of the pterygoid flange (Figs. 5A–5D). The medial pneumatic foramen is homologous to the ‘foramen pneumaticum’ of Baumel & Witmer (1993), and the ‘foramen pneumaticum basiorbitale’ of Elzanowski, Paul & Stidham (2001) and Elzanowski & Stidham (2010) for avian theropods.

Posterior pneumatic foramen (ppf)

Aperture or recess on the posterior surface of the quadrate body, typically at mid-height of the quadrate (Figs. 2G and 5). The posterior pneumatic foramen is similar to and likely homologous to the ‘foramen pneumaticum caudomediale’ of Elzanowski & Stidham (2010) for avian theropods (Fig. 1C).

Anterior pneumatic foramen (apf)

Aperture or recess on the anterior surface of the quadrate body, typically at mid-height of the quadrate (Fig. 5K). The anterior pneumatic foramen is likely homologous to the ‘foramen pneumaticum medial’ of Elzanowski, Paul & Stidham (2001), and the ‘foramen pneumaticum rostromediale’ of Elzanowski & Stidham (2010).

Ventral pneumatic foramen (vpf)

Aperture or recess on the ventral surface of the quadrate. The ventral pneumatic foramen is equivalent to the ‘foramen pneumaticum adventitium’ (or ‘ectopic pneumatic foramen’) of Elzanowski & Stidham (2010) for avian theropods (Figs. 5I and 5J).

Posterior pneumatic fossa (ppfo)

Shallow and well-delimited pneumatic recess on the posterior surface of the quadrate body, at mid-height of the bone and medial to the quadrate foramen (Fig. 5E).

Lateral pneumatic fossa (lpfo)

Shallow and poorly-delimited pneumatic recess on the ventral portion of the lateral surface of the quadrate, directly dorsal to the ectocondyle (Fig. 5L).

Morphological Variation in Quadrate Sub-units

To establish comparisons between taxa with a widely disparate quadrate morphology, a homology concept of the feature in question is required. Here, we will give a general account of the variability within different anatomical sub-units of the quadrate by following the criteria summarized in Rieppel (2006) to establish inter-taxic topological homologies.

The quadrate ridge is easily distinguishable in many theropod taxa such as Dilophosaurus wetherilli (Welles, 1984; Fig. 3C), Aerosteon riocoloradensis (MCNA-PV 3137; Fig. 4C) and Proceratosaurus bradleyi (NHM R.4860) but the demarcation of this structure may be only subtly developed, as in Noasaurus leali (PVL 4061), Majungasaurus crenatissimus (FMNH PR 2100; Fig. 3I), and Eustreptospondylus oxoniensis (OUMNH J.13558; Fig. 3U). The quadrate ridge is developed as a ‘columnar ridge’ in many theropod taxa such as Dilophosaurus wetherilli (Welles, 1984), Allosaurus ‘jimmadseni’ (SMA 0005; Allosaurus ‘jimmadseni’ sensu Chure, 2000; Loewen, 2010) and Eotyrannus lengi (MIWG 1997.550) but also forms a thin crest as in Tyrannosauridae (AMNH 5027; Carr , 1996; Brusatte, Carr & Norell, 2012). Although the ventral portion of the quadrate ridge is usually demarcated just above the entocondyle of the mandibular articulation, its dorsal termination is more variable. The dorsal termination can reach the quadrate head like in Acrocanthosaurus atokensis (NCSM 14345) or flatten at the mid-height of the quadrate such as in Albertosaurus sarcophagus (Currie, 2003: Fig. 10B). The quadrate ridge can be divided into two ridges by a deep groove as in Allosaurus fragilis (AMNH 600) and Allosaurus europaeus (ML 415). The quadrate ridge can also flare at the second dorsal third of the quadrate, and reappears slightly more dorsally, as observed in some derived Spinosauridae (Hendrickx, Araújo & Mateus, 2014). Likewise, the ventral portion of the quadrate ridge can also dichotomize into two crests separated by a concavity such as in the tyrannosaurids Albertosaurus sarcophagus, Daspletosaurus sp. (Currie, 2003: Figs. 10 and 28) and Tyrannosaurus rex (AMNH 5027).

Figure 4 Topological homologies in the non-avian averostran quadrate.

Left quadrate (A–F) of Aerosteon riocoloradensis (MCNA-PV-3137) in (A) anterior, (B) lateral, (C) posterior, (D) medial, (E) dorsal, and (F) ventral views (courtesy of Martin Ezcurra). Left quadrate (G–K) of Alioramus altai (IGM 100-1844) in (G) anterior, (H) lateral, (I) posterior, (J) medial, and (K) dorsal views (courtesy of Mike Ellison © AMNH). Right quadrate (L) of Qianzhousaurus sinensis (GM F10004-1; reversed) in ventral views (courtesy of Stephen Brusatte). Right quadrate (M–Q) of Falcarius utahensis (UMNH VP 14559; reversed) in (M) anterior, (N) lateral, (O) posterior, (P) medial, and (Q) ventral views (courtesy of Lindsay Zanno). Left quadrate (R–W) of Bambiraptor feinbergi (AMNH 30556) in (R) anterior, (S) lateral, (T) posterior, (U) medial, (V) dorsal, and (W) ventral views. Abbreviations: afq, anterior fossa; dqjc, dorsal quadratojugal contact; ecc, ectocondyle; enc, entocondyle; ics, intercondylar sulcus; lpfo, lateral pneumatic fossa; lpq, lateral process; mfq, medial fossa; mpf, medial pneumatic foramen; pfq, posterior fossa; ppf, posterior pneumatic foramen; pfl, pterygoid flange; qf, quadrate foramen; qh, quadrate head; qjp, quadratojugal process; qr, quadrate ridge; vpdq, ventral projection of the dorsal quadratojugal contact; vpf, ventral pneumatic foramen; vqjc, ventral quadratojugal contact; vsh, ventral shelf of the pterygoid flange.

The pterygoid flange (Fig. 2D, pfl) contacts the quadrate process of the pterygoid anteriorly or anteromedially, and sometimes other bones such as the epipterygoid in Herrerasaurus ischigualastensis (Sereno & Novas, 1994) and possibly Incisivosaurus (Balanoff et al., 2009), the basisphenoid and prootic in Erlikosaurus andrewsi (Clark, Perle & Norell, 1994; Lautenschlager et al., 2014), and the squamosal in Khaan mckennai (Balanoff & Norell, 2012). Although the pterygoid flange can be easily homologized between taxa, it may acquire subtrapezoidal, subtriangular, subrectangular and M-shaped outlines, or form a large semi-oval structure. The ventral limit of the flange can reach the mandibular condyles (e.g., Tyrannosaurus rex, Baryonyx walkeri; Fig. 3P) or get attached to the quadrate body far dorsal to the mandibular articulation (e.g., Majungasaurus crenatissimus; Fig. 3J). This structure can in some instances be divided into two ridges separated by a deep pneumatic foramen facing ventrally (e.g., Alioramus altai; Fig. 5I; Tyrannosaurus rex FMNH PR2081). In anterior view, the pterygoid flange can be straight and only projected anteriorly, as in the carcharodontosaurid Shaochilong maortuensis (Brusatte et al., 2010: Fig. 7A), or anteromedially recurved. The anteroventral margin of the pterygoid flange can either be straight, or medially and/or dorsally deflected, forming a horizontally oriented or dorsally inclined shelf-like structure here referred as the ventral shelf, as in Majungasaurus crenatissimus (FMNH PR 2100; Fig. 3G), Carnotaurus sastrei (MACN-CH 894) and Allosaurus fragilis (Madsen, 1976: plate 3d).

The medial fossa of the quadrate (Fig. 2D, mfq) is easily homologized between taxa as it is always situated on the pterygoid flange, typically on its ventromedial surface. This fossa is posteriorly delimited by the quadrate body in non-avian theropods and sometimes by the ventral shelf of the pterygoid flange. The medial fossa can be of variable depth (deep in Cryolophosaurus; FMNH PR1821; shallow in Eustreptospondylus; OUMNH J.13558), pneumatized (e.g., Falcarius; UMNH VP 14559; Fig. 4P), and situated in the ventralmost part of the pterygoid flange (e.g., Tsaagan; IGM 100-1015) or at mid-height of it and directly dorsal to a large pneumatic recess like in Mapusaurus roseae (MCF PVPH-108.102).

The posterior fossa of the quadrate (Fig. 2B, pfq) can be located either in between the quadrate and the quadratojugal, being confluent with the quadrate foramen (e.g., Mapusaurus; MCF PVPH-108.102), or in the middle of the quadrate shaft and between the quadrate ridge and the lateral limit of the quadrate shaft (e.g., ‘Syntarsus’; MNA V2623), Tsaagan (Norell et al., 2006), Majungasaurus (Sampson & Witmer, 2007; Fig. 3I). The posterior fossa can either be strongly ventrodorsally elongated like in the carcharodontosaurid Acrocanthosaurus (NCSM 14345), or form an oval concavity lateromedially wide (e.g., Majungasaurus; Sampson & Witmer, 2007). Similarly to the medial fossa, the posterior fossa can host a pneumatic foramen positioned dorsally (e.g., Sinornithomimus; IVPP V11797–10) or ventrally (e.g., Garudimimus; IGM 100-13) inside the fossa.

Due to the highly variable morphology of the quadrate foramen, this structure deserves special attention. It can be completely absent (e.g., Carnotaurus, Torvosaurus, Eustreptospondylus; Fig. 3U), or form a very small aperture (e.g., ‘Syntarsus’; Tykoski, 2005) to a large opening (e.g., Bambiraptor; Fig. 4T). In most non-avian theropods, only a small portion of the lateral margin of the quadrate foramen is delimited by the quadratojugal (e.g., Sinraptor; Currie, 2006) while in some non-avian theropods, the majority of the lateral margin is formed by the quadratojugal (e.g., Dromaeosaurus). Finally, in a few theropods, the foramen can be completely enclosed in the quadrate (e.g., Aerosteon; Sereno et al., 2008; Fig. 4C).

The quadratojugal contact of the quadrate (Fig. 2G, qjc) can either be a single extensive contact or made of two contacts separated by the quadrate foramen. In the latter case, the ventral quadratojugal contact and the dorsal quadratojugal contact of the quadrate are not always clearly separated and their dorsal and ventral margins, respectively, can overlap like in the sinraptorid Sinraptor dongi (IVPP 10600). If the quadrate foramen is absent or fully enclosed by the quadrate, the lateral quadratojugal contact typically forms an elongated line of variable width along the lateral margin of the quadrate. Where separated by the quadrate foramen, the ventral and dorsal contacts can display a wide variety of surface and outlines. Both quadratojugal contacts may face laterally, anteriorly or posteriorly, and their articulating surface can be smooth, irregular or deeply grooved by several radiating ridges, as in Allosaurus fragilis (Madsen, 1976). The ventral quadratojugal contact is typically D-shaped or ovoid in lateral view. Its anterior margin can extend far anteriorly, forming the quadratojugal process (Norell et al., 2006), and its ventral margin can project far laterally, as in Oviraptoridae (Maryańska & Osmólska, 1997). The dorsal quadratojugal contact varies from a very thin line to a broad surface in lateral or posterior views and its dorsal extension can reach the quadrate head or terminate well ventral to it. A ventral projection of this contact may be present, and such projection delimiting part of the lateral border of the quadrate can either be short, like in Daspletosaurus sp. (Currie, 2003: Fig. 28A) and Baryonyx walkeri (Fig. 3O), or form an elongated ramus, like in the therizinosaurid Falcarius utahensis (Zanno, 2010: Fig. 1H) and the basal coelurosaur Zuolong salleei (Choiniere et al., 2010: Fig. 3B).

In some basal theropods, ceratosaurs and dromaeosaurids, the lateral process of the quadrate (Fig. 2B, lpq) forms a wing-like projection similar to the pterygoid flange. This process is an extension of the quadrate body laterally so it is difficult to delimit. Such process is present in Allosaurus ‘jimmadseni’ (SMA 0005), Sinraptor dongi (Currie, 2006: Fig. 1D), and Erlikosaurus andrewsi (Clark, Perle & Norell, 1994: Fig. 7). The lateral process also varies in shape and size, as it can be lateromedially short and parabolic in posterior view (e.g., Carnotaurus; MACN-CH 894), or lateromedially elongated and subtriangular in posterolateral view (e.g., Dilophosaurus; UCMP 37302; Fig. 3B). Its ventral border can also extend to the quadrate foramen (e.g., Bambiraptor; AMNH 30556; Fig. 4T) or more ventrally, sometimes reaching the lateral condyle of the mandibular articulation (e.g., Ilokelesia, Majungasaurus; MCF PVPH 35, FMNH PR 2100; Fig. 3I).

The quadrate head always articulates with the cotylus of the squamosal and more rarely with other bones of the braincase such as the opisthotic in oviraptorids (Maryańska & Osmólska, 1997), the prootic in Mononykus olecranus (Perle, Chiappe & Barsbold, 1994; Chiappe, Norell & Clark, 2002) and the postorbital in Shuvuuia deserti (Chiappe, Norell & Clark, 1998; Chiappe, Norell & Clark, 2002). The contact between the quadrate and the opistothic-exoccipital/paroccipital process is also present in Herrerasaurus ischigualastensis (Sereno & Novas, 1994), Dilophosaurus wetherilli (Welles, 1984), Ceratosaurus magnicornis (Madsen & Welles, 2000; Sanders & Smith, 2005), tyrannosaurids (Currie, 2003), Heyuannia huangi (Lü, 2005), and Erlikosaurus andrewsi (Lautenschlager et al., 2014), yet this contact occurs on a small medial surface just immediately dorsal to the quadrate head and not with the quadrate head itself. The large majority of non-avian theropods have a monostylic quadrate head (Rauhut, 2003; C Hendrickx, pers. obs., 2015); yet, oviraptorids (Maryańska & Osmólska, 1997: Fig. 3B), the alvarezsaurid Shuvuuia deserti (Chiappe, Norell & Clark, 1998), and some tyrannosaurids such as Tyrannosaurus and Gorgosaurus (Larson, 2013) have the apomorphic condition of possessing a bistylic quadrate head. In those theropods, the otic capitulum of the quadrate head always contacts the braincase. This condition has also been observed in the dromaeosaurid Mahakala omnogovae (Turner, Hwang & Norell, 2007) but Turner, Pol & Norell (2011: Fig. 4) later reconsidered the head of the quadrate as not being bistylic. The morphology of the quadrate head is variable in dorsal view; it is subtriangular in most basal theropods (Sereno & Novas, 1994) like Dilophosaurus (UCMP 37302; Fig. 3E), Erlikosaurus (Lautenschlager et al., 2014) and Bambiraptor (AMNH 30556; Fig. 4V), oval to subcircular in megalosaurids like Afrovenator (UC OBA1) and Torvosaurus (BYUVP 9246), and allosauroids such as Aerosteon (MCNA-PV-3137; Fig. 4E), Sinraptor (IVPP 10600) and Shaochilong (IVPP V2885.3), or subquadrangular in some Spinosaurinae such as Irritator (SMNS 58022). While the dorsal surface of the quadrate head is either convex or flattened in posterior view in most non-avian theropods, the quadrate head of some allosaurids (Bakker, 1998: Fig. 5C) and derived tyrannosaurids (FMNH PR208) shows a well-marked concavity on the dorsal margin. The quadrate head can also be conical in posterior view, as in Oviraptoridae (Maryańska & Osmólska, 1997: Fig. 1B). Despite this variability, the quadrate head can be easily homologized inter-taxically due to the obvious location of this structure.

Figure 5 Morphology and position of pneumatic openings in the quadrate of non-avian Theropoda.

Right quadrate (A) of the carcharodontosaurid Acrocanthosaurus atokensis (NCSM 14345; reversed) in medial view. Left quadrate (B) of the carcharodontosaurid Mapusaurus roseae (MCF-PVPH-108) in medial view. Left quadrate (C) of the carcharodontosaurid Giganotosaurus carolinii (MUCPv CH 1) in medial view. Right quadrate (D) of the therizinosaur Falcarius utahensis (UMNH VP 14559; reversed) in medial view (courtesy of Lindsay Zanno). Right quadrate (E) of the metriacanthosaurid Sinraptor dongi (IVPP 10600; reversed) in posterior view (courtesy of Philip Currie). Left quadrate (F) of the neovenatorid Aerosteon riocoloradensis (MCNA PV 3137) in posterior view (courtesy of Martín Ezcurra). Left quadrate (G) of the ornithomimid Garudimimus brevipes (IGM 100–13) in posterior view (courtesy of Yoshitsugu Kobayashi). Right quadrate (H) of the dromaeosaurid Buitreraptor gonzalezorum (MPCA 245; reversed) in posterior view. Right quadrate (I) of the tyrannosaurid Alioramus altai (IGM 100–844) in ventral view (courtesy of Mick Ellison). Left quadrate (J) of the tyrannosaurid Tyrannosaurus rex (FMNH PR2081; cast, reversed) in ventral view. Left quadrate (K) of the carcharodontosaurid Mapusaurus roseae (MCF-PVPH-108) in anterior view. Left quadrate (L) of the neovenatorid Aerosteon riocoloradensis (MCNA PV 3137) in lateral view (courtesy of Martín Ezcurra). Abbreviations: apf, anterior pneumatic foramen; lpq, lateral process; lpfo, lateral pneumatic fossa; mpf, medial pneumatic foramen; ppf, posterior pneumatic foramen; ppfo, posterior pneumatic fossa; qf, quadrate foramen; vpf, ventral pneumatic foramen. Scale bars = 10 cm (A–C, J, K), 5 cm (E–G, L), 1 cm (D, H, I).

With the exception of the therizinosaur Erlikosaurus andrewsi and the ornithomimosaur Sinornithomimus dongi which both seem to have an autapomorphical tricondylar condition on the mandibular articulation (Clark, Perle & Norell, 1994; Kobayashi & Lü, 2003; Lautenschlager et al., 2014), all other non-avian theropods have two mandibular condyles. The presence of three mandibular condyles was also noted in the oviraptorosaur Avimimus portentosus (Chatterjee, 1995) and the dromaeosaurid Sinornithosaurus millenii (Xu & Wu, 2001). However, Vickers-Rich, Chiappe & Kurzanov (2002) only found two condyles in Avimimus and our observations confirm that the third condyle of Sinornithosaurus seems to be part of the much broader lateral condyle (Xu & Wu, 2001: Fig. 4D).

The intercondylar sulcus (Fig. 2E, ics) varies in orientation, size and depth. It can be large, shallow and sub-perpendicular to the long axis passing through the mandibular articulation as in Tyrannosaurus rex (FMNH PR2081), or narrow, deep and strongly lateromedially-oriented as in some derived spinosaurids (e.g., MHNM.KK376).

In posterior view, the shape of the mandibular articulation (Fig. 2C, mar) can vary from the biconvex condition known in most theropods, to the W-shaped articulation typical of Citipati osmolskae (Clark, Norell & Rowe, 2002: Fig. 6) or a single convex articulation seen in some dromaeosaurids such as Tsaagan mangas (IGM 100/1015). In Tsaagan, the convex outline of the mandibular articulation in posterior view results from a poor delimitation of the ecto- and entocondyle and the separation of these two condyles by a shallow intercondylar sulcus; yet this morphology might be due to the bad preservation of the mandibular condyle. A posterior intercondylar notch (Fig. 2J, pin) was observed in Allosaurus (Bakker, 1998: Fig. 5B, C; SMA 0005) and Suchomimus tenerensis (MNN GAD 502) whereas an anterior intercondylar notch (Fig. 2K, ain) is present in the abelisaurids Majungasaurus crenatissimus (FMNH PR 2100; Fig. 3L) and Carnotaurus sastrei (MACN-CH 894).

Pneumaticity of the quadrate can be externally expressed by pneumatic foramina or restricted to an internal chamber within the quadrate bone. The establishment of inter-taxic homologies is difficult to assess because these structures have very diverse interspecific variability. Nevertheless, as in other saurischian taxa (Schwarz, Frey & Meyer, 2007), these pneumatic structures have phylogenetic signal (e.g., Gold, Brusatte & Norell, 2013; Hendrickx, Araújo & Mateus, 2014; see below). These openings can appear on different sides and portions of the quadrate. The medial and posterior pneumatic foramina (Fig. 2G, ppf) usually occur in the medial and posterior fossa respectively, and their position inside these fossae is quite variable. Pneumatic foramina can also be located in a pneumatic recess outside the medial fossa and directly ventral to it such as in the carcharodontosaurids Mapusaurus roseae (Coria & Currie, 2006) and Acrocanthosaurus atokensis (Eddy & Clarke, 2011). In the latter, the pneumatic aperture is divided by a septum.

Figure 6 Distribution of quadrate pneumaticity in Theropoda.

Cladogram of non-avian theropods based on the theropod classification summarized by Hendrickx, Mateus & Araújo (2015) and showing the phylogenetic distribution of quadrate pneumaticity and the different quadrate pneumatic foramina in theropod dinosaurs. Silhouettes by Funkmonk (Dilophosaurus, Shuvuuia, Dromaeosauroides, and Suzhousaurus), M. Martyniuk (Ornitholestes and Similicaudipteryx), T. Michael Keesey (Deinocheirus), Choiniere et al. (2010; Zuolong; modified) and S. Hartman (all others).

Review of the Quadrate Function and Pneumaticity in Non-avian Theropods

Function of the quadrate

In all archosaurs, and all amniotes except Mammaliaformes, the main function of the quadrate is the articulation of the cranium with the mandible, yet this bone also plays an important role in the mobility of the skull in many extant theropods. Streptostyly is a fundamental property of all avian theropods, and quadrate kinesis in birds, known already in the beginning of the 19th century (Nitzsch, 1816), has been extensively studied over the past sixty years (e.g., Fisher, 1955; Bock, 1964; Bock, 1999; Bock, 2000; Bühler, 1981; Zusi, 1984; Zusi, 1993; Bühler , 1985; Bühler, Martin & Witmer, 1988; Chatterjee, 1991; Chatterjee, 1997; Hoese & Westneat, 1996; Zweers, Vanden Berge & Berkhoudt, 1997; Zweers & Vanden Berge, 1998; Bout & Zweers, 2001; Gussekloo & Bout, 2005; Meekangvan et al., 2006). Streptostyly consists of the rotation of the quadrate at its dorsal articulation against the squamosal which typically leads to a transverse movement, although a lateral movement of the quadrate around an anteroposteriorly directed axis occurs in some lepidosaur taxa (Metzger, 2002). Cranial kinesis in avian theropods with a streptostylic quadrate includes upward (protraction) and downward (retraction) rotation of the rostrum relative to the braincase. Three main types of kinesis, in which the role of the quadrate is relatively equivalent, are recognized relative to the position of the dorsal flexion zone of the cranium and the nature of the nasal opening in modern theropods (Bock, 1964; Bühler, 1981; Zusi, 1984; Meekangvan et al., 2006). In prokinesis, flexion occurs at the nasofrontal joint and the upper jaw thereby moves as one unit; in amphikinesis, flexion occurs in two zones of flexibility and the upper jaw and its tip are bent upward; in rhynchokinesis, flexion occurs forward from the nasofrontal joint, allowing its anterior part to be moved (Zusi, 1984).

Inference of the cranial kinesis and quadrate mobility in non-avian theropods has been investigated by Holliday & Witmer (2008) which regard the cranium of this group of dinosaurs as partially kinetically competent, because synovial joints and protractor muscles are present, but not fully kinetic like in birds. The strong suture of the quadrate to the quadratojugal and the immobile contact of the quadrate and the pterygoid on the medial side of the pterygoid flange in most non-avian theropods seem to indicate a limited movement, and perhaps even the total absence of movement within the cranium. Although the synovial quadrate head joint existing in theropods, and all other archosaurs, is necessary to infer cranial kinesis, its presence in akinetic taxa such as crocodiles demonstrates that the synovial joint cannot be considered alone as an argument for cranial kinesis. Synovial joints have actually been interpreted as growth zones rather than articular surfaces of mobile joints based on the presence of very thin articular cartilage covering the end of this joint (Holliday & Witmer, 2008). According to Holliday & Witmer (2008), “articular cartilage persists in loading environments that exert hydrostatic pressures (which result in a change in volume but not shape) but exert low shear stresses.” Indeed, one of the key centers of deformation during normal biting is the quadrate-squamosal contact, which would have experienced large shear stresses associated with torque and asymmetrical loading during biting (Rayfield, 2005), and the presence of a minimal amount of cartilage between the quadrate and squamosal would therefore suggest that the synovial zone was rather a growth zone than a mobile one. A streptostylic quadrate in Tyrannosaurus rex (Molnar, 1991; Molnar, 1998), Nanotyrannus lancensis (Larson, 2013), Oviraptor philoceratops (Smith, 1992), Heyuannia huangi (Lü, 2005) and Dromiceiomimus brevitertius (Russell, 1972) based on the saddle joint between the quadrate and squamosal only is therefore unlikely.

Nevertheless, and more convincingly, a streptostylic quadrate was also proposed in the alvarezsaurid Shuvuuia deserti by Chiappe, Norell & Clark (1998). In this taxon, the quadratojugal/jugal? (n.b., Dufeau (2003) considers the quadratojugal to be absent in Shuvuuia deserti), instead of being firmly sutured to the quadrate as in other non-avian theropods, would have contacted the lateral surface of the quadrate through a movable joint (Chiappe, Norell & Clark, 1998; Chiappe, Norell & Clark, 2002). According to Chiappe, Norell & Clark (1998), the absence of a laterodorsal contact of the quadrate with the quadratojugal/jugal, as well as a lateroventral process of the squamosal, would have permitted the quadrate to pivot anteroposteriorly, and the upper jaw to rotate ventrodorsally due to this transversal movement. These authors have implied the existence of a bending zone between the frontals and the nasal–preorbital bones in S. deserti, allowing the flexion of the snout as a single unit when the quadrate displaced forward, like in prokinetic birds. Nevertheless, the complex contacts between the nasal, frontal and prefrontal illustrated by Sereno (2001: Fig. 12B) makes assessment of Chiappe, Norell & Clark’s (1998) hypothesis dubious (Holliday & Witmer, 2008). Holliday & Witmer (2008) also note that the maxillojugal and palatal flexion zones necessary to allow a true prokinesis in alvarezsaurids are still not clearly defined. Likewise, the contact between the pterygoid flange of the quadrate and the pterygoid also needs to be better documented in order to imply any specific movement of the quadrate inside the cranium of S. deserti.

A movable articulation between the quadrate and quadratojugal was proposed in the oviraptosaurids Heyuannia huangi (Lü, 2003) and Nemegtomaia huangi (Lü et al., 2004; Lü et al., 2005). In Heyuannia, the quadrate and quadratojugal articulation forms a trochlea-like structure (Lü, 2003; Lü, 2005), while the quadratojugal contact of Nemegtomaia is diagnostically convex and was described as a lateral cotyle by Lü et al. (2004). Although such articulation suggests some mobility between the quadrate and quadratojugal, it is unlikely that the skull of these two oviraptorids could display avian-like kinesis. As in other non-avian theropods, the oviraptorid quadrate was an immovable bone (Barsbold, 1977; Maryańska & Osmólska, 1997) so that the quadratojugal, if kinetic, could only pivot either ventrodorsally or mediolaterally from the quadratojugal contact of the quadrate. Yet, the quadratojugal of at least Nemegtomaia does not seem to have a loose articulation with the jugal given that the articulating surface between the two bones is anteroposteriorly extensive (Lü et al., 2004: Fig. 2), disallowing mobility between the jugal and quadratojugal. Consequently, we consider unlikely that movement was possible between the quadrate and quadratojugal in Heyuannia and Nemegtomaia and, unlike Barsbold (1977), see the oviraptorosaur skull as akinetic.

Quadrate articulation with the mandible and orientation of the intercondylar sulcus are highly variable among non-avian theropods, therefore suggesting some variation in the movement of the mandibular rami when the jaw opened. The helical intercondylar sulcus present in many non-avian theropods (C Hendrickx, pers. obs., 2015) was noticed by Bakker (1998) in basal theropod dinosaurs, by Hendrickx & Buffetaut (2008) in spinosaurids, and by Molnar (1991) and Larson (2008) in Tyrannosaurus rex. These authors suggested that the spiral groove of the mandibular articulation constrained the diagonal ridge of the articular glenoid fossa, which fitted into the intercondylar sulcus, to slide laterally. This would force the mandibular rami of the mandible to displace laterally when the lower jaw was depressed, enlarging the width of the larynx in order to swallow large-size prey items (Hendrickx & Buffetaut, 2008).

In Allosaurus, the enlargement of the mandibular condyles associated with the posteroventral inclination of the ventral part of the quadrate, and the intercondylar notch, were interpreted by Bakker (1998) as joint-stabilization zones. According to Bakker (1998), the anteroposterior enlargement of the articulating surface would improve the stability of the mandibular articulation when the mouth was widely opened, whereas the intercondylar notch, morphologically convergent to the depression of knee joints in crocodiles and birds, would be hosting one or several ligaments within the quadrate-mandibular articulation (Bakker, 1998). An intercondylar notch is present in the abelisaurids Carnotaurus sastrei (MACN-CH 894) and Majungasaurus crenatissimus (FMNH PR 2100), and the spinosaurid Suchomimus tenerensis (MNN GAD 502), perhaps implying similar jaw mechanics of the mandibular articulation as in Allosaurus. Yet, Bakker’s (1998) jaw mechanics hypotheses based on the shape of the mandibular articulation and the presence of an intercondylar notch require further investigation with modern functional analysis methods such as FEA to be tested.

Pneumaticity in the quadrate

Pneumatization of the quadrate bone has long been recognized for its phylogenetic value (e.g., Gauthier, 1986; Holtz, 1998; Chiappe, 2001; Rauhut, 2003; Holtz, Molnar & Currie, 2004; Smith et al., 2007; Benson, 2010; Carrano, Benson & Sampson, 2012; Turner, Makovicky & Norell, 2012; Novas et al., 2013; Choiniere et al., 2014b). Pneumatic foramina of the quadrate are widespread among avetheropod clades (Gold, Brusatte & Norell, 2013; Fig. 4). The presence of one or several pneumatic foramina has indeed been recorded in carcharodontosaurids (e.g., Coria & Currie, 2006; Eddy & Clarke, 2011), megaraptorans (Sereno et al., 2008), tyrannosauroids (e.g., Molnar, 1991; Brochu, 2003; Currie, 2003; Xu et al., 2004; Witmer & Ridgely, 2010; Brusatte, Carr & Norell, 2012; Gold, Brusatte & Norell, 2013), compsognathids (Currie & Chen, 2001), alvarezsauroids (J Choiniere, pers. comm., 2014), therizinosaurs (Clark, Perle & Norell, 1994; Zanno, 2010), oviraptorids (e.g., Maryańska & Osmólska, 1997; Lü, 2003; Kundrát & Janáček, 2007; Balanoff & Norell, 2012), ornithomimosaurs (e.g., Witmer, 1997; Tahara & Larsson, 2011), dromaeosaurids (Makovicky, Apesteguía & Agnolín, 2005) and troodontids (Barsbold, Osmólska & Kurzanov, 1987; Currie & Zhao, 1993; Varricchio, 1997; Xu et al., 2002; Xu & Norell, 2004). An incipient development of a pneumatic recess, the posterior pneumatic fossa, also exists in the basal allosauroid Sinraptor dongi (Currie, 2006), suggesting that quadrate pneumaticity may be an avetheropod synapomorphy. Yet, external manifestation of quadrate pneumaticity only occurs in derived members of Allosauroidea, Tyrannosauroidea, and Ornithomimosauria, and an apneumatic quadrate exists in the basal members of each of these clades (i.e., Sinraptor and Allosaurus for Allosauroidea (Currie, 2006; C Hendrickx, pers. obs., 2011), Tanycolagreus and Proceratosaurus for Tyrannosauroidea (Carpenter, Miles & Cloward, 2005; Rauhut, Milner & Moore-Fay, 2010), and Nqwebasaurus for Ornithomimosauria; see Choiniere, Forster & de Klerk’s (2012) codings of their datamatrix). Pneumatic foramina have not been reported for any alvarezsauroid taxa, but are present in basalmost members of Therizinosauria, Oviraptorosauria and Paraves. This suggests that external quadrate pneumaticity occurred independently in several basal avetheropod clades and is a possibly synapomorphy of the clade Therizinosauria + Pennaraptora (Fig. 6).

The pneumatic opening is particularly large in some allosauroids such as Aerosteon riocoloradensis (Sereno et al., 2008; Fig. 5F) and Acrocanthosaurus atokensis. (Eddy & Clarke, 2011; Fig. 5A), and the therizinosaur Falcarius utahensis (Zanno, 2010; Fig. 5D). It, however, forms a small rounded or oval aperture lodged in the posterior fossa of the quadrate body in most avetheropods (Fig. 5). The posterior pneumatic foramen is the most common quadrate pneumatic aperture in non-avian theropods and is seen in many coelurosaur clades. For instance, it is present in the tyrannosauroid Dilong paradoxus (Xu & Norell, 2004), the compsognathid Sinosauropteryx prima (Currie & Chen, 2001: Fig. 3F), the ornithomimids Hexing qingyi (the ‘quadratic foramen’ of Liyong, Jun & Godefroit, 2012), Garudimimus brevipes (the ‘foramen’ of Kobayashi & Barsbold, 2005; Fig. 5G), Sinornithomimus dongi (the ‘quadratic foramen’ of Kobayashi & Lü, 2003) and Struthiomimus altus (AMNH 5339), the basal oviraptorosaur Incisivorosaurus gauthieri (Balanoff et al., 2009), the dromaeosaurid Buitreraptor gonzalezorum (Makovicky, Apesteguía & Agnolín, 2005; Fig. 5H), and the troodontids Mei long (Xu & Norell, 2004), Sinovenator changii (Xu et al., 2002) and possibly Gobivenator mongoliensis (Tsuihiji et al., 2014). The posterior pneumatic foramen is, in fact, incorrectly interpreted by several authors as the quadrate foramen in ornithomimosaurs (e.g., Kobayashi & Lü, 2003; Kobayashi & Barsbold, 2005; Choiniere, Forster & de Klerk, 2012). A genuine quadrate foramen between the quadrate and quadratojugal, as seen in the large majority of other theropods, is found in most (possibly all) ornithomimosaurs possessing a posterior pneumatic foramen (e.g., Garudimimus, Struthiomimus; Kobayashi & Barsbold, 2005; C Hendrickx, pers. obs., 2015). Tahara & Larsson (2011) wrote that “no obvious foramen or fossa was identified on the posterior surface of the quadrate” in Ornithomimus edmontonicus. Yet, a deep posterior fossa seems to be present on the right side of the specimen they studied (TMP 95-110-1; n.b., the fossa seems to be filled with sediment on the left side), in the homologous position of that of the posterior fossa of other ornithomimosaurs (C Hendrickx, pers. obs., 2015). It is, therefore, surprising that this fossa was apneumatic, as implied by Tahara & Larsson (2011). Consequently, we consider likely that a posterior pneumatic foramen was also leading to the pneumatic chamber hosting the quadrate diverticulum in this taxon. An incipient development of a posterior pneumatic foramen is seen in Sinraptor dongi in which the quadrate, though apneumatic, includes a well-delimited pneumatic fossa between the quadrate foramen and quadrate ridge (Currie, 2006; Fig. 5E). The presence of a posterior pneumatic foramen is a possible synapomorphy of the clade Pennaraptora, which encompasses Oviraptorosauria and Paraves (Foth, Tischlinger & Rauhut, 2014; Fig. 6). The medial pneumatic foramen, located in the ventral corner of the pterygoid flange, has also been reported in several theropod clades. It is present in the carcharodontosaurids Acrocanthosaurus atokensis (Eddy & Clarke, 2011; Fig. 5A), Mapusaurus roseae (Coria & Currie, 2006; Fig. 5B), and Giganotosaurus carolinii (MUCPv-CH-1; Fig. 5C), the tyrannosaurids Albertosaurus sarcophagus (Currie, 2003: Fig. 10B) and Tyrannosaurus rex (Molnar, 1991; Brochu, 2003), the therizinosaur Falcarius utahensis (Zanno, 2010; Fig. 5D), the oviraptosaurids Conchoraptor gracilis and possibly Ajancingenia yanshini (Maryańska & Osmólska, 1997; Kundrát & Janáček, 2007), and the basal avialan Archaeopteryx lithographica (Domínguez et al., 2004). A pneumatic foramen has also been noticed in the mediodorsal part of the quadrate in the ornithomimosaur Ornithomimus edmontonicus (Tahara & Larsson, 2011). A pneumatic foramen piercing the quadrate medially is a probable synapomorphic feature of Carcharodontosauridae or carcharodontosaurids more derived than Concavenator corcovatus and/or Eocarcharia dinops, pending on the results of the latest phylogenetic analyses on carcharodontosaurids (i.e., Ortega, Escaso & Sanz, 2010; Carrano, Benson & Sampson, 2012). In non-avian theropods, the ventral pneumatic foramen that occurs within a recess on the posteroventral part of the pterygoid flange (‘funnel-like external opening on the rostral surface of the quadrate, above the condyles’ of Gold, Brusatte & Norell, 2013: p. 37) is only present in Tyrannosauroidea. It is observed in the tyrannosaurids Alioramus altai (Brusatte, Carr & Norell, 2012; Gold, Brusatte & Norell, 2013; Fig. 5I), Daspletosaurus sp. (Currie, 2003: Fig. 28C) and Tyrannosaurus rex (Brochu, 2003; Witmer & Ridgely, 2010; Fig. 5J). In non-tyrannosaurid tyrannosauroids, such a ventral pneumatic foramen is present in Dilong paradoxus (Gold, Brusatte & Norell, 2013) but was not observed in the closely related taxa Guanlong wucaii, Proceratosaurus lengi, and Xiongguanlong baimoensis (Gold, Brusatte & Norell, 2013). It is also not clearly present in Eotyrannus lengi (contra Gold, Brusatte & Norell, 2013; C Hendrickx, pers. obs., 2011). A ventral pneumatic foramen of the quadrate is most likely synapomorphic of non-proceratosaurid Tyrannosauroidea (Fig. 6). A pneumatic foramen can also be seen on the anterior surface of the quadrate, as in Mapusaurus roseae (Coria & Currie, 2006; Fig. 5K), Heyuannia huangi (Lü, 2005), Erlikosaurus andrewsi (Lautenschlager et al., 2014), Troodon formosus (Currie & Zhao, 1993), and perhaps Tyrannosaurus rex (Molnar, 1991). More rarely, a pneumatic fossa can be situated on the lateral and posterior surface of the quadrate body, as in Aerosteon riocoloradensis (MCNA-PV 3137; Fig. 5L) and Sinraptor dongi (Currie, 2006; Fig. 5E), respectively. The presence of an anterior pneumatic foramen, a lateral pneumatic fossa, or a posterior pneumatic fossa is an autapomorphy in each of these taxa.

Carcharodontosauridae (Coria & Currie, 2006; Eddy & Clarke, 2011) and Tyrannosauridae (Molnar, 1991; Brochu, 2003) possess several pneumatic openings which perforate different sides of the quadrate and sometimes intercommunicate (Brochu, 2003). The pneumatic foramina usually enter a large pneumatic chamber within the quadrate bone as in Tyrannosaurus rex (Molnar, 1991; Brochu, 2003; Witmer & Ridgely, 2010), Alioramus altai (Gold, Brusatte & Norell, 2013), Conchoraptor gracilis (Kundrát & Janáček, 2007) or Ornithomimus edmontonicus (Tahara & Larsson, 2011). The neovenatorid Aerosteon riocoloradensis also possesses a large posterior pneumatic foramen leading to a pneumatic chamber (Sereno et al., 2008).

These pneumatic foramina and the pneumatic chamber associated with them are invaded by the quadrate diverticulum of the mandibular arch pneumatic system which, together with the periotic pneumatic system, forms the tympanic sinus of archosaurs (Dufeau, 2011; Tahara & Larsson, 2011). The mandibular arch pneumatic system includes the quadrate and/or the articular diverticulum which both have their embryological origins as parts of the first pharyngeal (= mandibular) arch, like the middle ear sac itself (Witmer, 1997). As in non-avian theropods, the quadrate diverticulum of modern birds exhibits a large variety of morphologies, and can pneumatize the quadrate by entering through a single medial or anteromedial foramen (Witmer, 1990; Tahara & Larsson, 2011). In basal theropods with an apneumatic quadrate, both medial and posterior fossae of the quadrate possibly represent the osteological trace of the quadrate diverticulum. In non-avian theropods having a pneumatic quadrate, the position of the quadrate diverticulum is variable as in ornithomimids (Tahara & Larsson, 2011), carcharodontosaurids and oviraptorids (C Hendrickx, pers. obs., 2015). The quadrate diverticulum of non-avian theropods may also have communicated with other diverticula such as the squamosal diverticulum as in Conchoraptor gracilis (Kundrát & Janáček, 2007), and the siphoneal diverticulum of the articular as in Dilong paradoxus, Aerosteon riocoloradensis and perhaps other non-avian maniraptorans (Sereno et al., 2008; Tahara & Larsson, 2011). In Tyrannosaurus rex, however, the siphoneal diverticulum does not pass through the quadrate, and the quadrate diverticulum only enters the ventral opening of the pterygoid flange, and then passes with or without the siphoneal diverticulum along the medial fossa of the pterygoid flange (Tahara & Larsson, 2011). Likewise, the quadrate diverticulum only pneumatizes two distinct regions of the quadrate in Acrocanthosaurus atokensis and Mapusaurus roseae (Tahara & Larsson, 2011).

Conclusions

Here we propose a revised nomenclature of the quadrate bone and a corresponding set of abbreviations that provide a standard set of terms for describing this cranial bone in non-avian theropod dinosaurs. The quadrate can be divided into five regional categories—the quadrate body, quadrate head, mandibular articulation, pterygoid flange, and pneumatic foramina and fossae—and many anatomical sub-units such as—the quadrate shaft, quadrate head, quadrate ridge, quadrate foramen, lateral process, quadratojugal contact, squamosal contact, pterygoid contact, mandibular articulation, medial fossa, and posterior fossa. Although they are highly variable in shape, all quadrate entities, with perhaps the exception of the posterior fossa, are easy to homologize across taxa, and a description of their morphology should be provided in the literature.

The quadrate of the large majority of non-avian theropods is akinetic, and it is unlikely that a streptostylic quadrate is present in the derived alvarezsauroids Shuvuuia deserti, as was previously thought. A lateral movement of the rami while the mandible was depressed occurred in various theropods (e.g., spinosaurids). This lateral movement of the rami was due to a helicoidal and diagonally oriented intercondylar sulcus of the mandibular articulation. The presence of an intercondylar notch in allosaurids is interpreted as a joint-stabilization zone that would improve the stability of the mandibular articulation when the mouth was widely opened. However, this assumption needs further investigation from modern functional morphology techniques.

A pneumatic quadrate is present in members of most non-avian avetheropod clades, in which a pneumatic foramen is seen in the ventral part of the pterygoid flange and in the medial and lateral fossae. Pneumatic foramina invading the quadrate seem to be independently acquired by allosauroids, tyrannosauroids, compsognathids, and ornithomimosaurs throughout their evolution. The presence of pneumatic foramina in the quadrate of basalmost members of therizinosaurs, oviraptorosaurs, troodontids and dromaeosaurids suggests that quadrate pneumaticity is a synapomorphy of the clade Therizinosauria + Pennaraptora. Although the pneumatic recess invaded by the quadrate diverticulum of the mandibular arch pneumatic system is linked to a single pneumatic foramen in most avetheropods, the presence of several pneumatic openings perforating different sides of the quadrate has been recorded in Carcharodontosauridae and Tyrannosauridae.

Supplemental Information

Supplemental Information 1 Supplemental Information

Function of quadrate sub-entities, quadrate sub-units terminology, and quadrate ontogeny in Lourinhanosaurus autunesi and Shuvuuia deserti.

Click here for additional data file.

We thank editor Andrew Farke (Raymond M. Alf Museum of Paleontology) and reviewers Jonah Choiniere (Uni. Witwatersrand) and Federico Agnolin (MACN) who kindly provided insightful comments that greatly improved this paper. The quadrate of many non-avian theropods were examined first hand in several institutions and we thank P Sereno (Uni. Chicago), P Makovicky (FMNH), W Simpson (FMNH), M Lamanna (CMNH), A Henrici (CMNH), M Carrano (NMNH), M Brett-Surman (NMNH), S Chapman (NHM), P Barrett (NHM), P Jeffery (OUMNH), S Hutt (MIW), R Allain (MNHN), R Schoch (SMNS), H-J Siber (SMA), C Dal Sasso (MSNM), A Kramarz (MACN), F Novas (MACN), R Barbieri (MPCA), L Salgado (MUCPv), J Ignacio Canale (MUCPv-CH), R Coria (MCF PVPH), C Succar (MCF PVPH), J Calvo (CePaLB), R Martínez (PVSJ), C Mehling (AMNH), M Norell (AMNH), D Krauze (SBU), J Groenke (SBU), P Brinkman (NCSM), and L Zanno (NCSM) for access to specimens in their care. Photographs of theropod quadrates were kindly shared by M Lamanna (CMNH), M Ezcurra (MACN), R Delcourt (Uni. São Paulo), M Carrano (USNM), E Buffetaut (CNRS), M Ellison (AMNH), L Witmer (Uni Ohio), S Brusatte (Uni. Edinburgh), R Benson (Uni. Oxford), C Foth (BSPG), P Currie (Uni. Alberta), J Canale (MUCPv-CH), P Barrett (NHM), J Choiniere (Uni. Witwatersrand), D Eddy (Uni. Texas), P Viscardi (Horniman), S Nesbitt (Uni. Texas), Y Kobayashi (HUM), R Tahara (McGill Uni.), R Pei (AMNH), C Dal Sasso (MSNM), P Sereno (Uni. Chicago), C Abraczinskas (Uni. Chicago), N Smith (Uni. Chicago), L Zanno (FMNH), R Tykoski (MNSD), D Burnham (Uni. Kansas), P Asaroff (MACN), R Irmis (UMNH), V Shneider (NCMNS), C Brochu (Uni. Iowa), S Lautenschlager (Uni. Bristol), M Mortimer, K Peyer (MNHN), and R Molnar (MNA), and the authors would like to address their sincere thanks to all of these people. We acknowledge the use of Phylopic for the theropod silhouettes, and thank Scott Hartman, Funkmonk, M Martyniuk, and T Michael Keesey for providing their artworks on Phylopic. A special thank goes to Paolo Viscardi for taking photos of the ostrich quadrate at the Horniman Museum & Gardens, and D Dufeau for sharing his MSc thesis on Shuvuuia. We also thank Isabel Torres for giving a final check in the English. CH dedicates this paper to the memory of Roger Bec.

Institutional Abbreviations

AMNH American Museum of Natural History, New York, USA

BHI Black Hills Institute, Hill City, South Dakota, USA

BYUVP Brigham Young University Vertebrate Paleontology, Provo, Utah, USA

CMNH Carnegie Museum, Pittsburgh, Pennsylvania, USA

FMNH Field Museum of Natural History, Chicago, Illinois, USA

GM Ganzhou Museum, Ganzhou City, Jiangxi Province, China

IGM Mongolian Institute of Geology, Ulaan Bataar, Mongolia

IVPP Institute for Vertebrate Paleontology and Paleoanthropology, Beijing, China

MACN Museo Argentino de Ciencias Naturales, Buenos Aires, Argentina

MCF PVPH Museo Municipal Carmen Funes, Paleontologia de Vertebrados, Plaza Huincul, Argentina

MCNA Museo de Ciencias Naturales y Antropológicas de Mendoza, Mendoza, Argentina

MIWG Dinosaur Isle, Isle of Wight Museum Services, Sandown, UK

ML Museu da Lourinhã, Lourinhã, Portugal

NCSM North Carolina Museum of Natural Sciences, Raleigh, North Carolina, USA

MNHN Muséum national d’Histoire Naturelle, Paris, France

MNA Museum of Northern Arizona, Flagstaff, Arizona, USA

MNN Musée National du Niger, Niamey, Niger

MPCA Museo Provincial Carlos Ameghino, Cipolletti, Río Negro, Argentina

MSNM Museo di Storia Naturale di Milano, Milan, Italy

MUCPv Museo de Ciencias Naturales de la Universidad Nacional de Comahue, Neuquén, Argentina

NH Horniman Museum & Gardens, London, UK

NHM The Natural History Museum, London, UK

OUMNH Oxford University Museum, Oxford, UK

PVL Fundación ‘Miguel Lillo,’ San Miguel de Tucumán, Argentina

PVSJ Instituto y Museo de Ciencias Naturales, San Juan, Argentina

SMA Sauriermuseum Aathal, Aathal, Switzerland

SMNS Staatliches Museum für Naturkunde, Stuttgart, Germany;

RTMP Royal Tyrrell Museum of Palaeontology, Drumheller, Alberta, Canada

UCMP University of California Museum of Paleontology, Berkeley, California, USA

UC University of Chicago Paleontological Collection, Chicago, USA;

UMNH Utah Museum of Natural History, Salt Lake City, Utah, USA

Additional Information and Declarations

Competing Interests

Author Contributions

The authors have declared that no competing interests exist.

Christophe Hendrickx conceived and designed the experiments, performed the experiments, analyzed the data, contributed reagents/materials/analysis tools, wrote the paper, prepared figures and/or tables, reviewed drafts of the paper.

Ricardo Araújo conceived and designed the experiments, performed the experiments, analyzed the data, contributed reagents/materials/analysis tools, wrote the paper, reviewed drafts of the paper.

Octávio Mateus conceived and designed the experiments.

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
