# Peer review of "The non-avian theropod quadrate I: standardized terminology with an overview of the anatomy and function"

_PeerJ, doi:10.7717/peerj.1245_

## Round 0.1 · original submission · Major Revisions

- The reviewers and I are in agreement that this is work is publishable, although some additional modifications to the text and figures are necessary prior to acceptance.

- Reviewer 1 states, and I agree, that the section on the ontogeny and taxonomy of Shuvuuia deserti seems out of place in this paper. Reviewer 2 has some similar concerns with the issue. I advise removing this from the current manuscript and breaking it into a separate paper.

- Reviewer 1 states that pneumatic characters should be incorporated into a species-level phylogenetic analysis; because this is done in the companion manuscript to the paper, there is no need to address the issue further (other than perhaps a citation to the other paper).

- The reviewers both provide numerous suggestions for improving the figures, particularly in the addition of labeling.

- Although the reviewers do not bring this up, I think it is important to give some consideration to the morphology and terminology of the quadrate in other archosaurs - particularly crocodilians (e.g., Holliday & Witmer 2008; Busbey 1989; or the brief consideration in Romer's Osteology of the Reptiles). How can this inform your own terminology?

- Because the paper addresses standardizing anatomical nomenclature, there should be at least some philosophical consideration of the appropriate issues of how to choose between competing terminologies, particularly within groups more or less closely related to extant taxa. These were discussed within the last decade within Harris 2004 (Anatomical Record A 281:1240-1246) and Wilson 2006 (JVP 26:511-518). When multiple terms have been used in the literature, how did you decide which to use?

- For Figure 5, please ensure that you have permission to include the silhouettes used here under a CC-BY publication license, and that the artists are attributed appropriately. Additionally, Bhullar et al. 2012 (in the Figure 5 caption) is not in the reference list.

- Both of the reviewers and I provide some recommendations for revising the phrasing and clarity of the text.

- The sections on ontogeny, pneumaticity, and functional morphology read more as literature reviews than as unique contributions. I recommend including some more synthesis of what the review means; or if this is already done, be more explicit about what your new contributions are on the topic.

·

Basic reporting

Dear Editor,
Here I am attaching the review of the manuscript entitled "The nonavian theropod quadrate I: standardized terminology and overview of the anatomy, function and ontogeny" authors C. Hendrickx, H. Araújo, and O. Mateus. The article contains a lot of novel information and constitutes an important contribution to the undertanding of quadrate morphology and terminology. Congratulations to the authors. I think that it must be published after minor revisions. Most comments were included in the attached pdf,
All the best,

Federico Agnolin,

Experimental design

No comments

Validity of the findings

No comments

Additional comments

No comments

·

Basic reporting

The English of the article needs to be improved - many sentences are unclear because of their structure or because they use incorrect terminology. I have highlighted many of these instances, but a careful read of the document must be done before it is resubmitted.

Most of the figures could be substantially improved. I have made notes in text detailing my suggestions. In summary, the current figure one should be moved to the text position of figure 2. Figure 2 should likewise be moved to the position of figure one. Both of these figures need substantial addition in terms of more views of the quadrate and substantial reorganization to make them clear to the reader. Figure 4 needs to include more labeling and more positional references. I have also suggested that the authors include a schematic figure showing the positions of quadrate pneumaticity in a generalized theropod quadrate and also present a phylogenetically arranged summary of how various theropod families express external signs of quadrate pneumatization. Omitting the coloring on the taxon names and silhouettes in Figure 5 would improve the clarity of the message.

This submission represents two units of publication: one paper on the terminology and homologies of various quadrate morphologies across Theropoda, and another on assessing the taxonomic identity of the "juvenile" Shuvuuia deserti specimen and the ontogeny of the quadrate in parvicursorine alvarezsauroids. I strongly suggest dividing the work along these lines.

Experimental design

The authors' inquiry is sound in design and relatively clear in its question.

Validity of the findings

Most of the authors' anatomical findings are robust. I think that the author's investigation of the ontogeny and taxonomy of Shuvuuia deserti, based on quadrate morphology, are incomplete and preliminary, and that they should fully develop those findings in a seaparate paper. Conclusions about the joint-stabilization zones in Allosauridae are really just restatements of Bakker and are not supported with additional analysis. Finally, the authors map the presence of quadrate pneumatization on a generalized theropod tree, but do not investigate the plesiomorphic condition of the quadrate at the base of major theropod clades, rendering their summary at best too coarse to be useful and at worst positively misleading about the evolution of this feature. I strongly suggest that the authors code pneumatic characters in a species-level phylogenetic analysis of theropods (e.g., borrowed from something like Choiniere et al 2014), and investigate the plesiomorphic condition for major theropod clades in a much finer-scale manner.

Additional comments

In general I think that this is a good piece of work, and an excellent means of standardizing theropod quadrate nomenclature. I think that the conclusions about Shuvuuia are best left for another paper, and that the writing needs to be edited very carefully so that your anatomical definitions are precise and consistent throughout. I have made numerous suggestions for how to improve your figures - I strongly suggest you consider them because many of the figures are unclear at the moment. I have attached a PDF file with numerous specific comments for your perusal. I have suggested "major revisions" because I believe that editing the figures and text will take a considerable amount of time, especially if you consider a more involved phylogenetic approach for investigating quadrate pneumaticity.

---

## Round 0.2 · Minor Revisions

Thank you for your close attention to the first round of reviews, and for your patience in awaiting the next round (I was delayed in finishing up the handling due to fieldwork).

A few minor suggestions have been raised by the reviewers; please address these in revision.

Finally: For the silhouettes in Figure 6, please confirm that all are licensed or can be licensed under a strict CC-BY license. CC-BY-SA-NC or CC-BY-NC or other variants cannot be used as PeerJ figures (which imposes a CC-BY license), unless you have explicit permission from the original artists. E.g., the Ornitholestes by M. Martyniuk is listed on PhyloPic as CC-BY-NC-SA. I see Mr. Martyniuk listed in the acknowledgments, but unless he explicitly cleared use of the silhoutte for release as CC-BY alone, you will need to find or create a different illustration. Sorry to keep hammering on this seemingly pedantic point, but it is important both for PeerJ as well as the artists involved to make sure that the proper licensing conventions are observed.

·

Basic reporting

Dear Editor PeerJ,

After a detailed review on the manuscript entitled "The non-avian theropod quadrate I: standardized terminology and overview of the anatomy, function and ontogeny" authors Christophe Hendrickx, Ricardo Araújo, and Octávio Mateus, I think that the manuscript should be accepted without any improvement or addition. The authors responded and modified accordingly most of the issues suggested by me and the other reviewed. Added novel information, and a large number of new figures. In this way, I think thta shoudl be accepted without new additions,
Everything else in which I can help you, please let me knwo,
All the best

Experimental design

No comments

Validity of the findings

No comments

Additional comments

No comments

·

Basic reporting

The English of the article has improved, but there are numerous areas of the text where clarification or more precise language is required. In the attached data file I highlight, explain, and often correct these areas, however the authors need to carefully revjse the text one more time.

The sections on the ontogeny of the quadrate (including SI) seem disjunct with the main goal of this paper, which is a standardized anatomical nomenclature. I urge the authors to consider whether it might make more sense to do a separate paper investigating the development of the quadrate in theropod dinosaurs (I personally feel it would be well-cited).

Several improvements to the figures could still be made. Figure 1 has labels for figure sections that are almost the same size as the bone images themselves, and lots of extraneous black space that could be more efficiently used. It would be helpful to see the images of the Struthio quadrate as large as possible on the page. Figure 2 shows great use of color to denote areas discussed in the article, however the line weights could be adjusted to show greater depth, and color could be used across the quadrate line drawings to standardize them and make comparison much easier.

There are some citations that are inappropriate, and these are flagged on the marked up pdf. There are other areas where observations about morphology of theropod quadrates are not cited or given appropriate specimen numbers. These stand out from the rest of the text because in other areas the authors have been extremely diligent to cite extensively.

Experimental design

no comments

Validity of the findings

no comments

Additional comments

Dear Authors,

The paper has made great progress since my first review, but it still needs many small clarifications and improvements before it is suitable for publication. My attached comments and changes should help guide your revisions. Please consider my comment about the ontogeny section carefully - I personally subscribe to a "one idea/one paper" concept, but acknowledge that there is room for debate about that philosophy. Please also note that I found a few minor errors in the supplemental information.

Sincerely,
Jonah

---

## Round 0.3 · accepted · Accept

Thank you for your close attention to the previous round of comments. In my view, the manuscript is ready for publication.